# Development of the AI Pipeline for Corneal Opacity Detection

**DOI:** 10.3390/bioengineering11030273

**Published:** 2024-03-12

**Authors:** Kenji Yoshitsugu, Eisuke Shimizu, Hiroki Nishimura, Rohan Khemlani, Shintaro Nakayama, Tadamasa Takemura

**Affiliations:** 1Graduate School of Information Science, University of Hyogo, Kobe Information Science Campus, Kobe 6500047, Japan; takemura@ai.u-hyogo.ac.jp; 2OUI Inc., Tokyo 1070062, Japan; hiroki@ouiinc.jp (H.N.); rohan@ouiinc.jp (R.K.); p.shintaro@ouiinc.jp (S.N.); 3Department of Ophthalmology, Keio University School of Medicine, Tokyo 1608582, Japan; 4Yokohama Keiai Eye Clinic, Kanagawa 2400065, Japan

**Keywords:** deep learning, semantic segmentation, corneal opacity detection, AI pipeline

## Abstract

Ophthalmological services face global inadequacies, especially in low- and middle-income countries, which are marked by a shortage of practitioners and equipment. This study employed a portable slit lamp microscope with video capabilities and cloud storage for more equitable global diagnostic resource distribution. To enhance accessibility and quality of care, this study targets corneal opacity, which is a global cause of blindness. This study has two purposes. The first is to detect corneal opacity from videos in which the anterior segment of the eye is captured. The other is to develop an AI pipeline to detect corneal opacities. First, we extracted image frames from videos and processed them using a convolutional neural network (CNN) model. Second, we manually annotated the images to extract only the corneal margins, adjusted the contrast with CLAHE, and processed them using the CNN model. Finally, we performed semantic segmentation of the cornea using annotated data. The results showed an accuracy of 0.8 for image frames and 0.96 for corneal margins. Dice and IoU achieved a score of 0.94 for semantic segmentation of the corneal margins. Although corneal opacity detection from video frames seemed challenging in the early stages of this study, manual annotation, corneal extraction, and CLAHE contrast adjustment significantly improved accuracy. The incorporation of manual annotation into the AI pipeline, through semantic segmentation, facilitated high accuracy in detecting corneal opacity.

## 1. Introduction

### 1.1. Shortage of Ophthalmologists in Developing Countries

Despite the global increase in ophthalmologists, there remains a significant shortage in developing countries [1]. This shortage is compounded by the limited access to appropriate surgical technologies and diagnostic tools [2,3]. The deployment of local ophthalmologists is considered a cost-effective solution. However, a need for more professionals in developing countries remains a challenge [4].

### 1.2. Device Used in This Study

The smart eye camera (SEC) [5] used in this study to photograph the anterior segment of the eye was invented and developed by an active ophthalmologist to solve the problems encountered in ophthalmology treatment in Japan and developing countries; it is an ophthalmic medical device that has been successfully put into practical use as a medical device. The SEC is a smartphone attachment that enables observation of various anterior segment structures of the eyes, including the eyelid, conjunctiva, cornea, anterior chamber, iris, lens, and anterior vitreous. This device mirrors the functionalities of the conventional slit lamp microscopy [6,7]. Furthermore, the SEC facilitates the preliminary estimation and identification of several anterior segment pathologies, such as cataracts [8], primary angle closure [9], allergic conjunctivitis [10], and dry eye disease [11,12]. Its integration with smartphone technology not only enhances accessibility but also potentially expands the scope of ophthalmologic diagnostics in various settings. An image-filing system with a dedicated application was also used to enable remote ophthalmology treatment. The development of the SEC has made it possible for anyone to perform eye examinations at any time, regardless of location. We are diagnosing videos of the anterior segment of the eye sent via the cloud, and we are conducting research and development to perform the diagnosis using AI to support ophthalmologists.

### 1.3. Application Example of Deep Learning to Ophthalmology

Deep learning has been applied in various ways to diagnose conditions that affect the anterior segment of the eye. Applications range from detecting angle-closure in anterior segment optical coherence tomography (AS-OCT) images to diagnosing dry eye disease (DED) and identifying peripheral anterior synechia (PAS). For instance, a deep learning system was developed for angle-closure detection in AS-OCT images, which surpassed previous methods by utilizing a multilevel deep network that captured subtle visual cues from the global anterior segment structure, local iris region, and anterior chamber angle (ACA) patch [13]. Another study evaluated a deep learning-based method to autonomously detect DED in AS-OCT images, which showed promising results compared to standard clinical dry eye tests [14]. Deep learning classifiers have also been used to measure peripheral anterior synechia based on swept-source optical coherence tomography (SS-OCT) images, demonstrating good diagnostic performance for gonioscopic angle closure and moderate performance for PAS detection [15]. In addition, deep learning classifiers have been developed to detect gonioscopic angle closure and primary angle closure disease (PACD) based on a fully automated analysis of AS-OCT images, showing effective detection capabilities [16]. Another study focused on the diagnostic performance of deep learning for predicting the plateau iris in patients with primary angle-closure disease using AS-OCT images, which revealed a high performance in predicting the plateau iris [17]. Finally, a deep learning model was developed for automated detection of eye laterality in anterior segment photographs, which achieved high accuracy and outperformed human experts [18]. In summary, deep learning has shown significant potential for the diagnosis of various anterior eye conditions, offering automated, accurate, and noninvasive methods that could enhance clinical evaluations and improve access to eye care in high-risk populations [13,14,15,16,17,18].

### 1.4. Potential Problems with Deep Learning: Eye Diseases

Deep learning models have shown significant promise in the field of ophthalmology, particularly for detecting and diagnosing ocular diseases. However, these models have several limitations that must be considered. One of the primary limitations of this study was the need for further testing and clinical validation. Although deep learning models have demonstrated high accuracy in the automated image analysis of fundus photographs and optical coherence tomography images, additional research is required to validate these technologies in clinical settings [19]. Another area for improvement is the lack of disease specificity and the public generalizability of the models. Despite the satisfactory performance reported in previous studies, most deep learning models developed for identifying systemic diseases based on ocular data lack the specificity required for individual diseases and still need to be generalizable to the broader public for real-world applications [20]. Furthermore, deep learning models can be computationally expensive, and deploying them on edge devices may pose a challenge. This is particularly relevant when considering the variety of available models and the potential need for a combination of models to solve a given task. The computational demands of these models may limit their practicality in certain clinical settings [21]. Lastly, while deep learning models can predict the development of diseases such as glaucoma with reasonable accuracy, they may miss certain cases, especially those with visual field abnormalities but not glaucomatous optic neuropathy. This indicates that although DL models are powerful tools, they may need to be able to fully replace the nuanced judgment of trained medical professionals [22]. In summary, while deep learning models hold great potential for revolutionizing the diagnosis of ocular diseases, their limitations in clinical validation, disease specificity, computational demands, and the potential to miss certain cases must be addressed before they can be fully integrated into clinical practice.

### 1.5. Corneal Opacity Detection Research Using Deep Learning

Research on detecting corneal opacity using AI has focused on developing and applying sophisticated algorithms and machine learning models to improve accuracy and objectivity in assessing corneal conditions. AI algorithms analyze images from iris recognition cameras to quantify corneal opacification objectively. This is particularly relevant for diseases such as mucopolysaccharidoses (MPS), where the current methods are subjective or difficult to standardize [23]. Machine learning techniques, including artificial neural networks (ANNs), adaptive neuro-fuzzy inference systems (ANFISs), and committee machines (CMs), are being investigated for their capabilities in classifying corneal images, identifying abnormalities, and enhancing the quality of confocal corneal images, achieving high accuracy and saving clinicians’ time [24]. AI applications in corneal topography have been reviewed, focusing on interpreting topographical maps for detecting corneal ectasias, where combined metrics from different devices could improve the AI model performance [25]. Deep learning models are being developed to screen candidates for refractive surgery by evaluating corneal tomographic scans, with the potential to outperform traditional methods and provide guidance to refractive surgeons [26]. Based on these studies, AI has been leveraged to enhance the detection and classification of corneal opacity and other abnormalities. These include using image analysis algorithms for objective quantification, machine learning for image classification and quality enhancement, and deep learning for screening in refractive surgery. These advancements aim to provide more accurate, reliable, and efficient tools for ophthalmologists to improve patient care.

### 1.6. Motivation for Study

Based on this research environment, we found that there has been almost no research on machine learning for detecting corneal opacity, which is a cause of blindness.

### 1.7. Purpose of Study

Therefore, in this study, we developed an AI pipeline to determine the presence of corneal opacity using anterior segment videos captured using a portable sitting microscope and deep learning techniques.

## 2. Materials and Methods

### 2.1. Ethical Approval

This study was conducted in strict accordance with the principles of the Declaration of Helsinki. Ethical approval for the study protocol was obtained from the Institutional Ethics Review Board of the Minamiaoyama Eye Clinic, Tokyo, Japan (IRB U. 15000127. Approval no. 202101). Owing to the retrospective design of the study and the use of anonymized data, the board waived the requirement for written informed consent from the participants.

### 2.2. Device to Capture the Anterior Segment Videos

Anterior segment videos were captured using a portable slit lamp microscope (Smart Eye Camera; SEC. SLM-i07/SLM-i08SE, OUI, Inc., Tokyo, Japan; 13B2X10198030 101/ 13B2X10198030201) (Figure 1). By attaching this device to a smartphone, it will be possible to perform eye examinations in the same way as with existing slit lamp microscopes. There is evidence that they do not require battery replacement or charging, are easy to carry, and exhibit the same performance and safety as existing medical devices in several regions [6,7].

### 2.3. Method to Build the Dataset

Data acquisition for this study was centralized at a singular ophthalmological facility, namely, the Yokohama Keiai Eye Clinic. The recordings, systematically obtained between July 2020 and December 2021, were subsequently collated on a dedicated cloud server, thus constituting the dataset for this study. The recording process entailed skilled ophthalmologists directing SEC toward the cornea of the patients, leveraging the device’s feature of emitting a white diffused light to facilitate clear visualization. The video capture protocol was aligned with the methodologies conventionally associated with slit lamp microscopes, ensuring standardization of the visual data. To further emulate the conditions of routine clinical assessments, patients were advised to avoid blinking during the video recording phase to enhance the consistency and clinical relevance of the collected video data.

### 2.4. Deep Learning for Corneal Opacity Detection

First, 30 diffuse light videos were decomposed into images; the images that could be used as validation data were selected. We used a program of our creation to divide the video into images. A total of 5996 images, 1617 positive frames, and 4379 negative frames were used to detect corneal opacity. The resolution of all images was 1280 pixels horizontally and 720 pixels vertically. Using these verified images, we attempted to detect corneal opacity by performing image classification. EfficientNet-B4 [27] was used as the convolutional neural network (CNN) model, and cross-validation was performed, but almost no correct positive frames could be detected. We chose CNN and Vision Transformer for image classification because they have been used frequently recently and were appropriate for comparison. We limited the impact of randomness on accuracy by fixing the random seed value throughout this study.

### 2.5. Improved Deep Learning for Corneal Opacity Detection

Therefore, to improve the detection accuracy, we manually annotated the cornea, which is the region of interest (ROI), extracted an image of only the cornea, adjusted the contrast using contrast-limited adaptive histogram equalization (CLAHE) [28], and used a convolutional neural network. Image classification was performed again using EfficientNet-B4 as a (CNN) model. The data structures used for the learning and prediction are presented in Table 1. The proportion of underlying diseases in the data is shown in the pie chart in Figure 2, the cornea extraction procedure is shown in Figure 3, and the contrast changes before and after CLAHE image processing are shown in Figure 4.

### 2.6. Hyperparameters for Training

Throughout this study, every hyperparameter for each model was determined by repeated 5-fold cross-validation. The hyperparameters at training were 30 for the number of epochs, 8 for the batch size, and 0.0001 for the learning rate, with default values for EfficientNet-B4 for the rest. Data augmentation during training included resizing (512 × 512), flipping up and down with a 1/2 probability, and flipping left and right with a 1/2 probability.

### 2.7. Semantic Segmentation

This confirmed that corneal opacity could be detected in the anterior segment images. However, to incorporate it into an AI-automated prediction system, which we call an AI pipeline, it is necessary to automate the manual annotation of the process of extracting the cornea and ROI from the anterior eye image.

Therefore, we performed semantic segmentation learning to segment the cornea from the anterior segment image by reusing the annotation mask used to extract the cornea from the anterior segment image and the original extracted image as learning data. U-Net [29]/EfficientNet-B4 was adopted as the semantic segmentation model.

### 2.8. Hyperparameters for Semantic Segmentation

The hyperparameters for learning the semantic segmentation were 40 epochs and 10 batch sizes. The learning rate started at 0.001 and decreased to 0.0001 after 25 epochs. All other values were set to the U-Net/EfficientNet-B4 default.

### 2.9. Data Augmentation Methods for Semantic Segmentation

Data augmentation during training included resizing (256 × 256), flipping left/right with 1/2 probability, Affine transformation, padding the edges according to image size, cropping at random, Gaussian noise with a probability of 1/5, and perspective transformation with a probability of 1/2. In addition, one set of augmentations among the following three was performed with a probability of 9/10: the first set includes CLAHE, brightness adjustment, and gamma transformation, the second set includes sharpening, blur, and motion blur, third set includes contrast adjustment and hue, saturation, and luminance change.

### 2.10. Environment for Study

This study was conducted on a Windows 11 system with the following specifications: CPU: Intel Core i7-11700KF, memory: 128GB, and GPU: RTX 4070.

In this way, an AI pipeline was completed that uses semantic segmentation to extract the ROI and cornea from anterior segment images and uses deep learning to classify images to detect corneal opacity.

## 3. Results

Table 2 shows the results of manually annotating the cornea as a region of interest (ROI), extracting only the cornea, adjusting the contrast with CLAHE, and learning with CNN (EfficientNet-B4).

Table 3 lists the metrics derived from the outcomes predicted by the model. The evaluation yielded commendable results across several key indicators: sensitivity, specificity, accuracy, and the area under the curve (AUC). The values obtained were as follows: sensitivity of 0.96 (95% confidence interval [CI]: 0.97–0.99), specificity of 0.96 (95% CI: 0.97–0.99), accuracy of 0.96 (95% CI: 0.97–0.99), and an AUC of 0.98 (95% CI: 0.98–0.99).

Figure 5 depicts the receiver operating characteristic (ROC) curve, illustrating the diagnostic ability of the model across various threshold settings.

Table 4 shows the outcomes of the corneal semantic segmentation, as predicted by the model. The Dice coefficient, also referred to as the F1 score, had a substantial value of 0.94. Furthermore, the intersection over union (IoU), another critical metric for segmentation performance, similarly registered a notable value of 0.94.

The Dice coefficient is called the “Sørensen-Dice index” or the “Sørensen-Dice coefficient”. The Dice coefficient DSC(A,B) for set A, and set B is defined by the following equation:(1)$$DSC\left(A,B\right)=\frac{2|A\cap B|}{\left|A|+|B\right|}$$

The Dice coefficient represents the ratio of the average number of elements in the two sets to the number of elements they have in common and is a value between zero and one. The larger the Dice coefficient, the more similar the two sets are.

The intersection over union (IoU) is an evaluation metric used in object detection and represents the percentage of image overlap. Specifically, it has a maximum value of 1 when the detected and true areas completely overlap and a minimum value of zero when there is no overlap at all. The IoU for regions A and B can be calculated using the following formula:(2)$$IoU=\frac{\left|A\cap B\right|}{\left|A\cup B\right|}$$

## 4. Discussions

We believe that these three approaches contributed to the improved prediction accuracy for corneal opacity. First, the cornea, the ROI, was extracted from the anterior segment image; second, the image resolution was reduced by changing the input image from the entire anterior segment image to the corneal image, the ROI, thereby reducing the reduction in image features; and third, CLAHE was applied as a contrast optimization.

It was also a good idea to reuse the anterior segment image used in the model training phase of corneal opacity prediction and the mask image used to extract the cornea to perform corneal semantic segmentation. The corneal semantic segmentation model eliminates the need to manually extract the cornea and allows it to be integrated into the AI pipeline. The corneal opacity prediction AI pipeline begins with the selection of anterior segment image frames from the video that were deemed suitable for diagnosis, followed by the extraction of the cornea through semantic segmentation, resulting in an accurate diagnosis of corneal opacity.

Despite the constraints presented by the limited size of the sampling dataset (comprising 5996 frames, with 1617 positive and 4379 negative frames), this study successfully developed a model with high diagnostic accuracy for corneal opacity. It is noteworthy that previous research in the domain of ocular image analysis often utilized datasets that exceeded thousands of annotated cases [30,31]. Conversely, the current study leveraged video data as the primary raw material, capitalizing on the potential to extract multiple image frames from a single video sequence. This methodology aligns with the techniques employed in prior research focused on the development of automated diagnostic AI systems [32,33], wherein methods such as cropping, flipping, and other forms of data augmentation are utilized to effectively expand the dataset from a single image. The implementation of these techniques, particularly the strategic use of video data for frame extraction and image amplification [34], is posited as a pivotal factor contributing to the development of a high-performance model despite the relatively modest size of the dataset.

In the context of developing diagnostic AI programs for medical applications, determining the optimal performance benchmarks, particularly for diagnostic goals, presents a substantial challenge. This is exemplified in the realm of ophthalmology, where certain diseases are the leading causes of blindness globally. Previous investigations, including our own, have underscored the potential of AI to achieve diagnostic accuracies comparable to, if not surpassing, those of human specialists. For instance, our prior research indicated that AI-based diagnostics could achieve over 95% accuracy in comparison with evaluations conducted by ophthalmologists in the context of a disease with a significant worldwide blindness burden [30]. Furthermore, Hu et al. reported an impressive diagnostic accuracy of 93.5% with an AUC of 0.9198 [35], indicating a high level of diagnostic precision. Additionally, a cross-sectional study by Son et al. demonstrated AI’s robust diagnostic performance, with an accuracy of 90.26% and an AUC of 0.9465 [36], further evidencing AI’s capability to accurately diagnose medical conditions. Moreover, recent studies provide compelling evidence of the efficacy of AI algorithms in distinguishing between infectious keratitis and immunological keratitis through image analysis. A notable report highlights the exceptional performance of the AI algorithm, as evidenced by AUC values of 0.986 for infectious keratitis and 0.960 for immunological keratitis [30]. These findings underscore the algorithm’s broad applicability not only in the identification of keratitis subtypes, but also in its performance across a range of ocular conditions, including corneal scars, ocular surface tumors, corneal deposits, acute angle-closure glaucoma, cataracts, and bullous keratopathy [30]. The deployment of this technology in ophthalmology clinics for professional use signifies a significant advancement in the field. It enables healthcare providers to more accurately identify the underlying causes of ocular diseases, thereby facilitating the determination of appropriate differential treatment methods. This development represents a pivotal step toward integrating AI into clinical practice, offering a promising tool for enhancing diagnostic accuracy and improving patient outcomes in ophthalmology. These findings collectively suggest that high diagnostic accuracy should be a key consideration in establishing performance benchmarks for AI systems aimed at diagnosing corneal opacity. Such evidence supports the argument for setting ambitious yet achievable accuracy goals in the development and evaluation of AI diagnostics, thereby enhancing their utility and reliability in clinical settings.

In the existing literature, there is a scarcity of studies employing deep learning methodologies for the identification of corneal opacity through images acquired via slit lamp microscopy. Consequently, this study is pioneering in its endeavor to develop a highly accurate model for the detection of corneal opacity. Moreover, the application of AI to the diagnosis of ocular pathologies from medical examination videos remains a nascent field. This research, therefore, holds significance because of its innovative approach to both the development of a precision model for corneal opacity detection and its exploration of AI-based diagnostic methodologies in ophthalmology. In addition, research on using AI to diagnose eye diseases from medical examination videos is new, and we believe that this research is significant in these two respects.

Describe practical considerations and challenges for implementing AI pipelines, including computational resources, data privacy, and integration with the existing healthcare infrastructure. A PC equipped with a GPU is sufficient for computing resources, and the data are video recorded on a smartphone, so no personal information is recorded. Integration with existing medical infrastructure can be easily achieved via LAN.

In addition, we discuss the potential clinical effectiveness, cost-effectiveness, and scalability of the solution proposed in this study. The solution proposed in this study is that it can be introduced at a much lower cost than expensive medical equipment, even in environments with poor ophthalmology treatment infrastructure, and that future research will allow the detection of other anterior segment diseases AI can easily be added.

Lastly, we discuss the clinical feasibility and acceptability of the AI pipeline proposed in this study. The devices used in this research have already been introduced in clinical settings worldwide, and an environment is already in place to diagnose anterior segment videos received via the cloud using the latest AI.

### Limitations of This Study

The current study had several limitations. First, the limited scale of the sample size was small. Despite the retrospective nature of the study, wherein the use of video recordings served to augment the dataset, the scope of the data remained relatively constrained. To develop robust and adaptable AI models, particularly those pertinent to imaging analysis, there is a significant need for more extensive datasets. Therefore, the limited sample size in this study may represent a significant impediment to the generalizability and comprehensive applicability of the derived AI models. In addressing the aforementioned limitation, this study drew inspiration from prior literature, which demonstrated enhanced detection of keratitis through the augmentation of single anterior segment images [34]. Such augmentation involves a six-fold increase in data quantity achieved by methods such as flipping, rotating, and cropping [34]. Similarly, our approach involved the meticulous recording of digital anterior segment videos, thereby facilitating amplification of the volume of raw data [32,33]. Second, the dataset in this study was exclusively sourced from a single medical institution, which may limit the external validity and generalizability of the findings. To ensure broader clinical applicability and substantiate the robustness of the conclusions, it is essential for future research endeavors to incorporate and validate the models against external datasets. Ideally, this validation process should involve a large-scale cohort comprising data from multiple medical facilities. This comprehensive approach will be instrumental in enhancing the reliability and relevance of AI models in diverse clinical settings.

## 5. Conclusions

We enhanced the accuracy of the prediction model by isolating the cornea, our region of interest (ROI), from anterior eye images. This refined the ability of the model to identify corneal opacity. Moreover, we streamlined the process by implementing semantic segmentation on the original and masked images instead of manual cornea extraction. Our AI pipeline for corneal opacity detection seamlessly extracts stills from videos, applies semantic segmentation for cornea extraction, and determines opacity presence.

Expanding beyond corneal opacity by integrating different disease detection modules, we aimed to create a versatile anterior segment diagnosis AI pipeline. This advancement could expedite screening during health checkups, lessening the ophthalmologists’ workload. To further enhance our corneal opacity detection AI pipeline, we aim to address three key areas: augmenting training data, refining image selection from video frames, and distinguishing between the right and left eyes in diagnostic images.

We plan to continue this research to complete an anterior segment diagnostic AI pipeline.

## Figures and Tables

**Figure 1 bioengineering-11-00273-f001:**
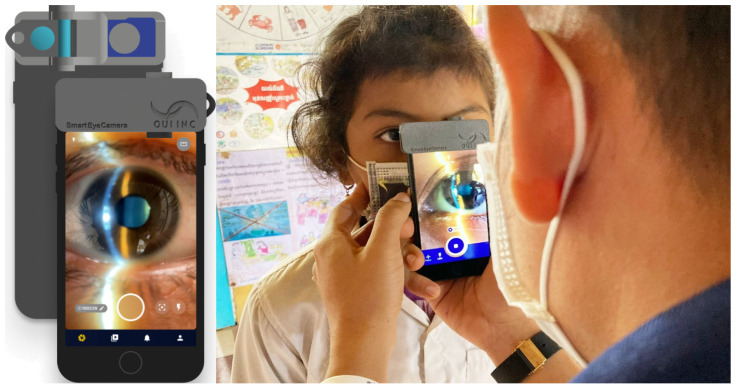
Smart eye camera.

**Figure 2 bioengineering-11-00273-f002:**
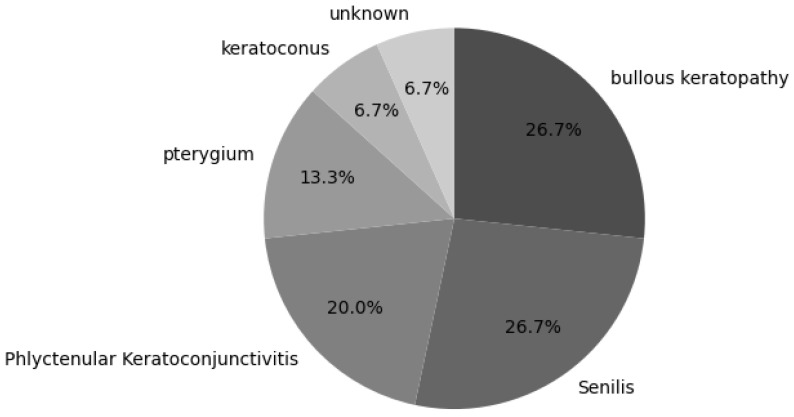
Ratio of underlying diseases. Bullous keratopathy and senilis account for the majority of the underlying diseases.

**Figure 3 bioengineering-11-00273-f003:**
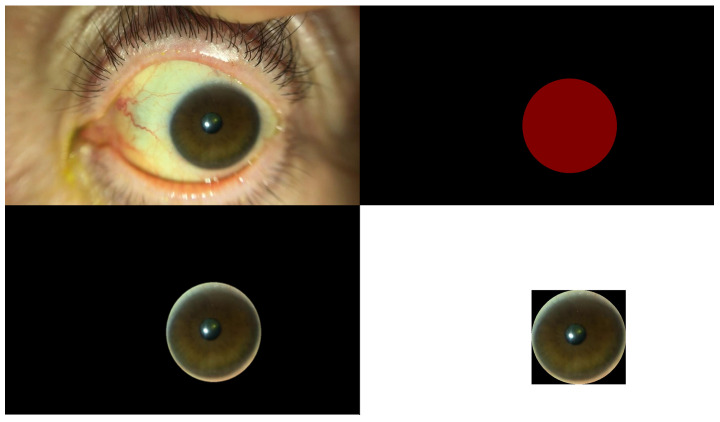
The upper left panel shows the original image frame extracted from the movie. The upper right panel shows the annotated mask of the cornea. The lower left corner is the cornea extracted using an annotated mask. The lower right corner is the extracted cornea (ROI only) and is an input image for training. The size of the input image is smaller than that of the original image.

**Figure 4 bioengineering-11-00273-f004:**
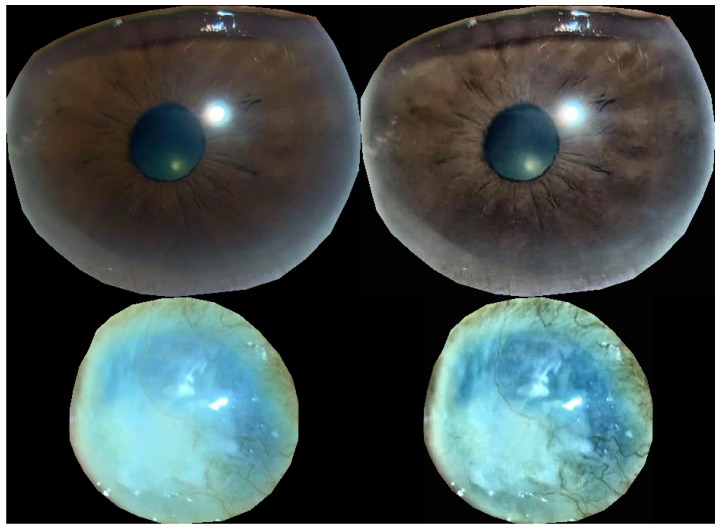
The upper left panel shows the extracted cornea image. The upper right is after CLAHE processing of the left image. The lower left panel shows the other extracted cornea image. The lower right is after CLAHE processing of the left image. It can be observed that the contrast of both images was improved.

**Figure 5 bioengineering-11-00273-f005:**
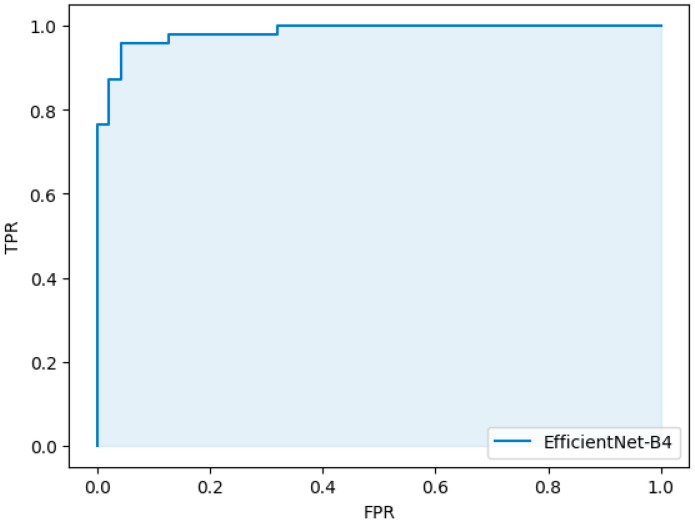
Receiver operating characteristic curve for prediction.

**Table 1 bioengineering-11-00273-t001:** Data structure.

	Negative	Positive	Total
train/val	188	188	376
test	47	47	94

**Table 2 bioengineering-11-00273-t002:** Confusion matrix.

Confusion Matrix	Value
True Positive	45
False Negative	2
False Positive	2
True Negative	45

**Table 3 bioengineering-11-00273-t003:** Performance of the model.

Evaluation Index	Value
Sensitivity	0.96 (95% CI. 0.97–0.99)
Specificity	0.96 (95% CI. 0.97–0.99)
Accuracy	0.96 (95% CI. 0.97–0.99)
AUC	0.98 (95% CI. 0.98–0.99)

**Table 4 bioengineering-11-00273-t004:** Dice and IoU of semantic segmentation.

Evaluation Index	Value
dice	0.94
IoU	0.94

## Data Availability

The data that support the findings of this study are available from the corresponding author upon reasonable request.

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
