# Peer review of "Development of the AI Pipeline for Corneal Opacity Detection"

_bioengineering, 2024, doi:10.3390/bioengineering11030273_

Round 1
Reviewer 1 Report
Comments and Suggestions for Authors
Find below some comments/feedback:
- The keywords should be between 4 - 6 words
- There is a need to briefly introduce the topic in the abstract (at least 2 sentences) and the paper's objective/aims should be clearly stated in the abstract
- Lines 40 - 110 contain a very long paragraph that isn't easy to comprehend. Kindly consider breaking it down accurately
- Before lines 112 - 114, kindly insert the motivation/research gaps of the study before the purpose of the study
- The materials & methods are too long (Line 115 - 180). consider having suitable subheadings
- Figures 2, 3, and 4 are not mentioned in the paper before their appearance
- Some paragraphs are too short. such as Lines 18-22; 153 - 156; 157 - 160; 161 - 164; 175 - 176; 177 - 179; 207 - 211
- Based on the plagiarism report, kindly recheck the content of the paper
- The conclusion needs to be reframed and concise
Comments on the Quality of English LanguageNeed minor editing and grammatical checking
Author Response
Thank you for your review of our manuscript.
>- The keywords should be between 4 - 6 words
Updated to 4 keywords.
Please confirm the revised version.
>- There is a need to briefly introduce the topic in the abstract (at least 2 sentences) and the paper's objective/aims should be clearly stated in the abstract
Updated.
Please confirm the revised version.
>- Lines 40 - 110 contain a very long paragraph that isn't easy to comprehend. Kindly consider breaking it down accurately
Divided it into three paragraphs.
Please confirm the revised version.
>- Before lines 112 - 114, kindly insert the motivation/research gaps of the study before the purpose of the study
Updated.
Please confirm the revised version.
>- The materials & methods are too long (Line 115 - 180). consider having suitable subheadings
Updated.
Please confirm the revised version.
>- Figures 2, 3, and 4 are not mentioned in the paper before their appearance
Updated.
Please confirm the revised version.
>- Some paragraphs are too short. such as Lines 18-22; 153 - 156; 157 - 160; 161 - 164; 175 - 176; 177 - 179; 207 - 211
Updated by providing subsections.
Please confirm the revised version.
>- Based on the plagiarism report, kindly recheck the content of the paper
We have checked our manuscript for plagiarism using Grammarly.
>- The conclusion needs to be reframed and concise
Updated.
Please confirm the revised version.

Reviewer 2 Report
Comments and Suggestions for Authors
The paper presents a approach to enhance accessibility and quality of ophthalmological care by using an AI pipeline to detect corneal opacity. The study utilized a portable slit-lamp microscope with video capabilities and cloud storage, combined with Convolutional Neural Network (CNN) models for image analysis and semantic segmentation. This research addresses the global inadequacies in ophthalmological services, especially in low- and middle-income countries, and contributes to the field by offering a cost-effective and efficient solution.
The paper is well-organized in general. I have following comments/suggestsions to improve it:
1. Provide a detailed comparison with existing methods for corneal opacity detection, highlighting the advantages and limitations of the proposed AI pipeline.
2. Conduct additional experiments and validation studies to assess the performance of the AI pipeline on diverse datasets and ensure its reliability in real-world clinical settings.
3. Discuss in more depth the practical considerations and challenges of implementing the AI pipeline, including computational resources, data privacy, and integration with existing healthcare infrastructure.
4. Expand the discussion section to include the potential clinical impact, cost-effectiveness, and scalability of the proposed solution.
5. Consider collaborating with ophthalmologists or medical professionals to validate the findings and provide insights into the clinical feasibility and acceptability of the AI pipeline.
Comments on the Quality of English LanguageNA
Author Response
Thank you for your review of our manuscript.
>1. Provide a detailed comparison with existing methods for corneal opacity detection, highlighting the advantages and limitations of the proposed AI pipeline.
Updated.
Please confirm the revised version.
>2. Conduct additional experiments and validation studies to assess the performance of the AI pipeline on diverse datasets and ensure its reliability in real-world clinical settings.
Currently, this study does not have sufficient data to perform additional experiments, and future work will include evaluating the performance of the AI pipeline on various datasets and ensuring reliability in real-world clinical settings.
>3. Discuss in more depth the practical considerations and challenges of implementing the AI pipeline, including computational resources, data privacy, and integration with existing healthcare infrastructure.
Updated.
Please confirm the revised version.
>4. Expand the discussion section to include the potential clinical impact, cost-effectiveness, and scalability of the proposed solution.
Updated.
Please confirm the revised version.
>5. Consider collaborating with ophthalmologists or medical professionals to validate the findings and provide insights into the clinical feasibility and acceptability of the AI pipeline.
Updated.
Please confirm the revised version.

Reviewer 3 Report
Comments and Suggestions for Authors
Dear Authors,
Article is well structured and the topic is interesting. However, following comments should be addressed prior to further processing of the article.
1) Refer to whole article: Why authors have selected CNN whereas other neural networks do exist?
2) Refer to line # 19: Check space in developing countries[1]. There should be a space between countries and [1]. Same has also been observed at other places as well. Recheck whole article for such typos.
3) Refer to line # 138: Which tool was used for decomposition of video into images?
4) Refer to line # 143: Weights assignment at the edges of CNN is usually random which may affect its accuracy. How do the authors deal with it? Authors may consider following study for study in this regard. https://www.mdpi.com/2071-1050/14/23/16317
5) Refer to line # 165: Good details regarding simulation setup are provided however did authors validate their model other than 40 epochs and 10 batch sizes?
6) Refer to line # 222: How did authors compare results accuracy and declared high diagnostic accuracy?
7) Refer to line # 233: It is better to mention limitations in a separate sub-section to make them prominent.
Good luck.
Comments on the Quality of English LanguageDear Authors,
Article is well structured and the topic is interesting. However, following comments should be addressed prior to further processing of the article.
1) Refer to whole article: Why authors have selected CNN whereas other neural networks do exist?
2) Refer to line # 19: Check space in developing countries[1]. There should be a space between countries and [1]. Same has also been observed at other places as well. Recheck whole article for such typos.
3) Refer to line # 138: Which tool was used for decomposition of video into images?
4) Refer to line # 143: Weights assignment at the edges of CNN is usually random which may affect its accuracy. How do the authors deal with it? Authors may consider following study for study in this regard. https://www.mdpi.com/2071-1050/14/23/16317
5) Refer to line # 165: Good details regarding simulation setup are provided however did authors validate their model other than 40 epochs and 10 batch sizes?
6) Refer to line # 222: How did authors compare results accuracy and declared high diagnostic accuracy?
7) Refer to line # 233: It is better to mention limitations in a separate sub-section to make them prominent.
Good luck.
Author Response
Thank you for your review of our manuscript.
>1) Refer to whole article: Why authors have selected CNN whereas other neural networks do exist?
We chose CNN and VisionTransformer for image classification because they have been used frequently recently and were appropriate for comparison.
I've updated, so please confirm the revised version.
>2) Refer to line # 19: Check space in developing countries[1]. There should be a space between countries and [1]. Same has also been observed at other places as well. Recheck whole article for such typos.
I've updated, so please confirm the revised version.
>3) Refer to line # 138: Which tool was used for decomposition of video into images?
We used a program of our creation to divide the video into images.
I've updated, so please confirm the revised version.
>4) Refer to line # 143: Weights assignment at the edges of CNN is usually random which may affect its accuracy.
How do the authors deal with it? Authors may consider following study for study in this regard. https://www.mdpi.com/2071-1050/14/23/16317
We limited the impact of randomness on accuracy by fixing the random seed value throughout this study.
I've updated, so please confirm the revised version.
>5) Refer to line # 165: Good details regarding simulation setup are provided however did authors validate their model other than 40 epochs and 10 batch sizes?
Throughout this study, every hyperparameter for each model was determined by repeated 5-fold cross-validation.
I've updated, so please confirm the revised version.
>6) Refer to line # 222: How did authors compare results accuracy and declared high diagnostic accuracy?
We compared the results of this study with those of related studies.
>7) Refer to line # 233: It is better to mention limitations in a separate sub-section to make them prominent.
We moved limitations to the last subsection of the Discussion section,
so please confirm the revised version.

Round 2
Reviewer 1 Report
Comments and Suggestions for Authors
The authors have addressed all the comments/suggestions made before. Only need to check some grammatical errors in the text.
I recommend the manuscript be considered for acceptance
Comments on the Quality of English LanguageMinor checking still require
Reviewer 2 Report
Comments and Suggestions for Authors
All issues addressed.
Reviewer 3 Report
Comments and Suggestions for Authors
Dear Authors,
My review comments are satisfactorily addressed.
Good luck.